# MAC: A Multimodal Benchmark for Understanding and Generating Academic Journal Covers

## Abstract

We introduce the Multimodal Academic Cover (MAC) benchmark to address the challenges of Large Multimodal Models (LMMs) in understanding and generating academic journal covers. While LMMs have demonstrated significant progress in creative arts and everyday applications, their capabilities in comprehending complex academic visuals and narratives remain underexplored. MAC comprises a collection of 5,872 cover images, accompanying cover stories, and associated articles from 40 prominent academic journals, providing a rich dataset for evaluation. We design bidirectional generative tasks—Image2Text and Text2Imag to assess authenticity and creativity in generating cover images and stories. Current LMMs, including DALL·E 3, GPT-4V, Gemini, CogView-3, GLM-4V, LLaVA, LLaMA-adapter, and MiniGPT4, are evaluated on this benchmark. Furthermore, we propose Multimodal Agent Linkage (MAL), a novel method to enhance conceptual comprehension within a long-context window. In-context learning techniques, such as few-shot learning, are also explored to improve the effectiveness of LMMs. All benchmarks, prompts, and codes will be released publicly.

## 1 Introduction

Emergent opportunities have occurred with the advent of Large Multimodal Models (LMMs) (Radford et al., 2021; OpenAI, 2023a;b). LMMs have revolutionized the interaction with integrated visual and textual content, impacting sectors ranging from creative arts to daily applications. They have been argued to achieve expert levels of such scenarios, *e.g.*, painting images based on text (Morris et al., 2023). However, their ability to comprehend and create scientific content remains a question.

Researchers have explored evaluating and improving scientific comprehension of large language models (LLMs). Galactica (Taylor et al., 2022), for instance, trains on a vast array of scientific materials, enabling it to effectively store, combine, and reason about scientific knowledge. Additionally, SCITUNE (Horawalavithana et al., 2023) introduces a tuning framework to enhance AI's understanding of complex scientific instructions across different modes. Similarly, Sci-CoT (Ma et al., 2023) proposes a two-stage framework that separates the processes of generating explanations and inferring answers, thus making better use of explanations during answer inference.

However, it's worth noting that prior works primarily focus on generating textual responses to scientific queries, often overlooking the visual aspect of scientific communication. Just as language plays a crucial role in conveying complex ideas, visual representations are equally important for deepening understanding. Visuals can distill and clarify abstract scientific concepts, making them more accessible and intuitive. For example, in the case of the Theory of Relativity, the visualization of the train experiment enhances comprehension by vividly demonstrating complex ideas like relative motion, time dilation, and length contraction. These visualizations enable a more intuitive grasp of the theory's principles, which might be challenging to convey through text alone.

In this paper, we attempt to see how far the LMMs are from human experts by benchmarking the most advanced LMMs in *multimodal scientific concepts*. In particular, we evaluate the capabilities of LMMs to understand and generate *academic journal covers*. Epitomized by publications like Cell (Cell, 2023), Nature (Nature, 2023), and Science (Science, 2023), the covers provide not only a visual and textual summary but also a window into the depth and context of the research they

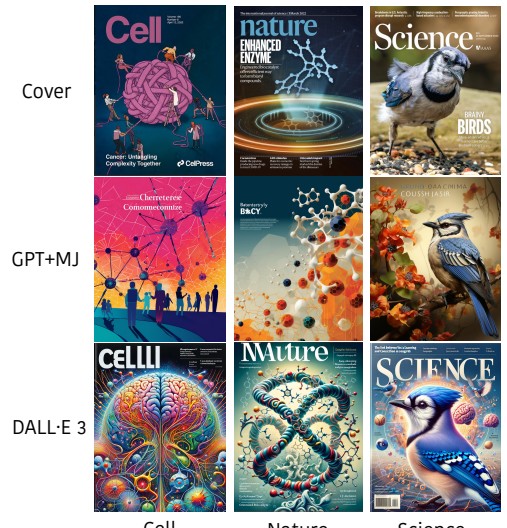

Cover

GPT+MJ

DALL·E 3

Cell       Nature       Science

### Cover Story

... Joining together the individual molecular components to form these compounds can be a challenge. In this weeks issue, Alison Narayan and her colleagues report the design and engineering of ==enzyme biocatalysts== that help to connect the necessary building blocks through their carbonhydrogen bonds. The researchers used engineered forms of ==cytochrome P450 enzymes== as the catalyst, and as a result were able to transform a low-yielding, unselective reaction into one that efficiently joined phenols to form biaryl compounds with control over both which ==CH bonds== are coupled and the stereochemistry of the products...

### GPT-4-Vision

Our latest issue's cover art encapsulates the essence of this scientific milestone, with futuristic influences and an intriguing blend of hatecore aesthetics that mirror the relentless pursuit of precision in ==biocatalysis==. Dive into the detailed world of science-based innovation, illuminated by high-key lighting, and digitally enhanced by the enigmatic power of nature-enhanced ==enzymes==. Within these pages, we explore the remarkable breakthrough in biocatalytic oxidative cross-coupling, led by ==cytochrome P450 enzymes==, addressing fundamental challenges in ==CC bond== formation.

Figure 1: **We propose the MAC benchmark to evaluate the visual understanding of academic journal covers for large multimodal models.** *Left* displays Different cover generations across three reputable journals in Text2Image. *Right* provides the comparison between a cover story and one written by GPT-4V in Image2Text, highlighting the ==correspondence== and ==discrepancies== .

encompass. Even for professional humans, creating academic covers and stories is quite challenging since it requires extensive cross-disciplinary knowledge, high-level abstraction, unified understanding across multiple modalities, and creativeness containing both literature and arts. Such capability is key for AI to advance toward human-level intelligence. Therefore, academic journal covers can support an investigation of whether LMMs can develop an understanding of abstract concepts akin to humans, thereby shedding light on the current disparity between AI and human intelligence.

To this end, we introduce the Multimodal Academic Cover (MAC) benchmark, a comprehensive collection sourced from eminent academic journals including Cell, Nature, Science, and their subsidiary publications.[1] MAC consists of 5872 journal issues, each with a complete group of cover images, stories, and relevant articles. As new scientific research continuously emerges and humanity's exploration of the unknown never ceases, our benchmark collection will keep growing and including cutting-edge research findings, ensuring that the benchmark remains challenging for emerging LMMs. Two more challenging subsets are also provided, MAC-Recent (940 issues) and MAC-Latest (50 issues), split by the publication time. Furthermore, we propose two bidirectional generative tasks, Image2Text and Text2Image, to assess the proficiency of LMMs in capturing and conveying complex scientific concepts in a contextually relevant manner.

MAC emphasizes dual-sided multimodal generation, driven by painting cover images and writing cover stories surrounding each journal issue. Previous multimodal research mainly concentrates on visual generation such as abstract concept depiction (Liao et al., 2023), or textual generation like question-answering tasks (Goyal et al., 2017; Chen et al., 2015; Hudson & Manning, 2019; Marino et al., 2019; Lu et al., 2022), neglecting the bidirectional generation of both images and texts, as well as their comparison. Nevertheless, a holistic understanding from the bidirectional view is a cornerstone of artificial general intelligence (Fei et al., 2022; Jain, 2023).

Our evaluation involves current state-of-the-art LMMs like GPT-4V (OpenAI, 2023b) and Gemini (Anil et al., 2023), including automatic evaluation agents and human experts. Three types of rating reference standards are considered, alignment of Text with the original Text (T-T), alignment of Image with the original Text (I-T), and alignment of Text with the original Image (T-I). We have conducted both qualitative (Figure 2) and quantitative experiments to demonstrate that evaluation agents are closely aligned with human preference and can accurately reflect the evaluated models.

---

[1]See license in license of Nature, license of Science, and license of Cell.

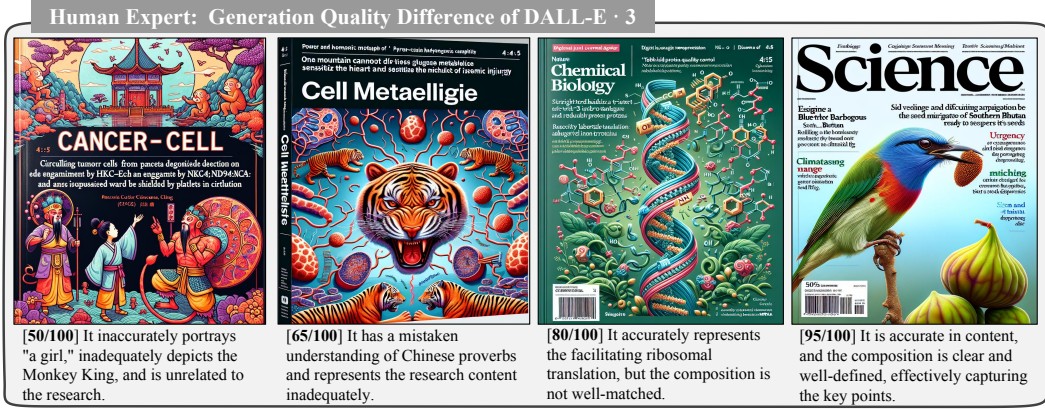

Figure 2: **DALL·E 3 (OpenAI, 2023a) displays different generation abilities when handling the Text2Image task.** Four distinct sample covers generated by DALL·E 3 are presented, each exemplifying the varying levels of quality produced by the model. Accompanying these covers are expert analyses detailing the rationale behind their assigned scores.

Experiments reveal the disability of LLMs to generate academic content and highlight the overlooked comprehension imbalance of academic disciplines for LMMs.

We observe that LMMs such as MidJourney (Midjourney, 2023) and Stable-Diffusion (Rombach et al., 2021) suffer from weak language capability facing long-context language windows. Therefore, we propose the Multimodal Agent Linkage, MAL, to combine Large Language Models like ChatGPT (OpenAI, 2023) and generative models like MidJourney (Midjourney, 2023) for a better multimodal understanding of scientific concepts. MAL simplifies the original long-context windows using LLM to provide a more suitable prompt for generative models. We also explore the potential of in-context learning techniques like few-shot learning (Brown et al., 2020) and find prompting methods fail in our cases. Experimental results show that MAL can improve the performance of LMMs when tackling challenging tasks. They also illustrate that the application of prompting techniques does not necessarily mean the improved capabilities of LMMs

**Our contributions:**

- We propose the MAC benchmark, a comprehensive collection containing cover images, cover stories, and relevant articles, sourced from leading academic journals including Cell, Nature, Science, and their sub-journals.
- We design bidirectional generative tasks for LMMs surrounding one journal issue, concentrating on the dual-sided understanding of academic covers during generation.
- We build Multimodal Agent Linkage that synergizes large language models with LMMs to enhance their understanding of scientific concepts in the long-context window.
- Our experimental results highlight the poor performance of current LMMs and emphasize disparities in capabilities across different topics, pinpointing the significance of multi-directional generative benchmarks.

## 2 RELATED WORK

**Multimodal Dataset for Evaluation** has emerged as a captivating avenue for researchers. Within this domain, a significant branch flourishes, encompassing question-answer datasets designed to tackle scientific problems. Noteworthy examples include AI2D (Kembhavi et al., 2016), FOOD-WEDS (Krishnamurthy et al., 2016), DVQA (Kafle et al., 2018), VLQA (Sampat et al., 2020), ScienceQA (Lu et al., 2022), and MathVista (Lu et al., 2023). Taking a comprehensive and multi-disciplinary approach, SciEval (Sun et al., 2023) is a benchmark for evaluating the scientific research abilities of models. In addition, Hessel et al. (Hessel et al., 2022) propose a benchmark for humor understanding that involves a multimodal task, shedding light on the mechanisms behind what makes a caption humorous. HallusionBench (Liu et al., 2023a) stands as another significant addition to

the landscape, serving as an image-context reasoning benchmark to explore language hallucination and visual illusion, a challenge that even GPT-4V (OpenAI, 2023b) and LLaVA-1.5 (Liu et al., 2023b). Meanwhile, LPM (Lee et al., 2023) has been introduced as a large-scale benchmark that scrutinizes the capabilities of vision-and-language models in comprehending educational videos through a multimodal lens. In contrast to the aforementioned datasets, which predominantly focus on text generation, our MAC represents a distinctive bidirectional generative benchmark. MAC delves into the realm of academic concept understanding, a formidable challenge that demands a profound repository of cross-disciplinary knowledge, high-level abstract reasoning skills, and a holistic grasp of information across multiple modalities, including both text and image generation.

**Scientific Understanding through Text Generation** has become a focal point for evaluating and enhancing the scientific comprehension capabilities of Large Multimodal Models (LMMs). Notable contributions in this space include Galactica (Taylor et al., 2022) which is trained on an extensive corpus comprising scientific papers, reference materials, knowledge bases, and various other sources. This training enables Galactica to effectively store, amalgamate, and reason about scientific knowledge. Additionally, SCITUNE (Horawalavithana et al., 2023) introduces a tuning framework designed to enhance the capacity of LMMs to follow complex scientific multimodal instructions. Meanwhile, Sci-CoT (Ma et al., 2023) presents a two-stage framework that segregates the processes of generating rationales and inferring answers, thereby facilitating a more efficient utilization of rationales during the answer inference phase. It is important to note that previous works primarily concentrate on addressing specific scientific queries from the perspective of textual generation. In contrast, our MAC tackles the more intricate task of comprehending academic concepts through bidirectional multimodal generative tasks, encompassing both text and image generation.

**Evaluation on Multimodal Benchmark** has Traditional evaluation metrics such as accuracy, BLEU (Papineni et al., 2002), CIDEr (Vedantam et al., 2015), IS (Salimans et al., 2016), FID (Heusel et al., 2017), and CLIP-Score (Hessel et al., 2021), take an exact match between the prediction and the target, which cannot evaluate objectively cutting-edge large multimodal models (LMMs) nowadays. Recent studies, such as mPLUG-Owl (Ye et al., 2023a;b) and LVLM-eHub (Xu et al., 2023) propose the human-involved subjective evaluation by incorporating human judgment. More works take ChatGPT-involved evaluation, such as MMBench (Liu et al., 2023d), CLAIR (Chan et al., 2023), and DEsignBench (Lin et al., 2023), which is destined to become increasingly widespread due to its automated nature. This paper takes human-involved and Large Model-involved evaluations to showcase state-of-the-art LMMs. Through our experiments, we further highlight the differences between existing Large Model-involved evaluations and human values.

Table 1: **MAC collects a full set of 5872 issues, including Nature, Science, Cell and their sub-journals.** Nature accounts for a significant proportion, and the main journals of CNS hold a high share due to their high quality. We have collected data dating back to 2006 for the earliest issues, aiming to cover as much data as possible. The average impact factor is calculated by summing up the impact factors of all journals of one journal family and then dividing by the total number of journals.

| Journal | #Issues | #Sub-journal | #Sub-journal Issues | Avg. Impact Factor | Start Year |
|---------|---------|--------------|---------------------|--------------------|-----------| 
| Cell | 319 | 843 | 7 | 38.4 | 2010 |
| Science | 265 | 1340 | 6 | 25.5 | 2009 |
| Nature | 547 | 2558 | 27 | 36.6 | 2006 |

## 3 DATASET AND TASK SETUP

MAC encompasses a comprehensive collection of prestigious academic journals, specifically targeting the families of Science (Science, 2023), Nature (Nature, 2023), and Cell (Cell, 2023), along with their respective sub-journals. This extensive compilation includes 40 distinct journals, ensuring the inclusion of every accessible issue. Each issue within the MAC database is systematically organized into three primary components: (1) Journal Cover, which provides the graphical representation of a specific issue's front cover; (2) Cover Story, offering a detailed narrative and introduction pertinent

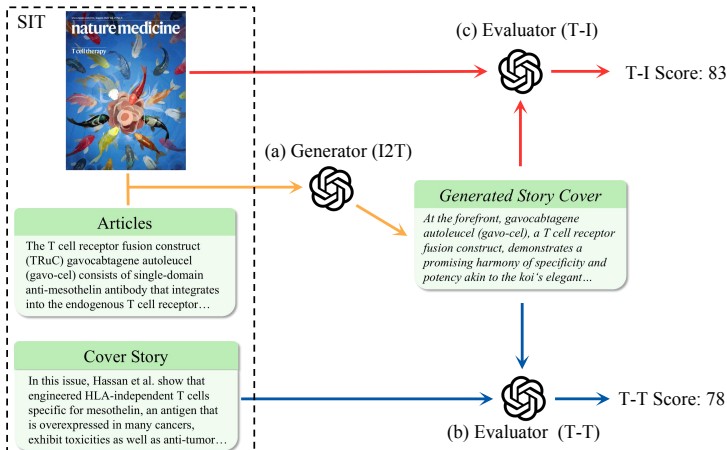

Figure 3: **The Image2Text task is to generate cover stories given cover images and articles.** (a) The Generator, powered by LMMs or our proposed Multimodal Agent Linkage, creates a cover story from the provided article and cover. Evaluations follow two paths: (b) Evaluators score the generated cover story against the original cover story for a *T-T score*; (c) they compare the original cover and generated cover story to assign a *T-I score*.

to the principal article of the issue; and (3) Cover Article[2], presenting a concise overview of the highlighted cover article.

## 3.1 MAC DATASET

The compilation of issues within MAC derives from the official websites of three eminent journal families. The details are in Table 1 and the brief introduction is shown as follows[3].

- **Cell (Cell, 2023) and its sub-journals**: MAC incorporates 7 journals related to the Cell publication series, mainly focusing on molecular and cell biology, with a total of 1162 issues.
- **Nature (Nature, 2023) and its sub-journals**: This segment encompasses 27 journals from the Nature publication family, totaling 3105 issues. Nature, renowned for its multidisciplinary approach, operates principal editorial offices across the United States, Europe, and Asia, featuring covers in high resolution.
- **Science (Science, 2023) and its sub-journals**: This segment includes 6 journals under the Science publication umbrella, emphasizing significant original academic research and comprehensive reviews.

MAC is an exceptionally high-quality dataset derived from top-tier academic journals like Science, Cell, and Nature. These journals maintain research credibility and feature aesthetically pleasing cover designs closely tied to their articles. Besides, the dataset is challenging due to its cutting-edge scientific content, often expressed in complex language. Furthermore, MAC is continuously updated with the latest academic achievements from these journals, offering a unique opportunity to test rapid learning instead of memorizing capabilities for Large Multimodal Models (LMMs).

## 3.2 MAC-LATEST AND MAC-RECENT

MAC-Latest and MAC-Recent are additionally introduced for different difficulty levels. Given that subsequent research is based on the foundation laid by prior studies, the ensuing increase in informational density and complexity inherently necessitates a deeper understanding of scientific

---

[2]For simplicity, we take the abstract of the cover article in this paper. Besides, we use the term *cover article* though a few issues may have more than 1 relevant article.

[3]The introduction is adapted from Wikipedia (Cell), Wikipedia (Nature), and Wikipedia (Science).

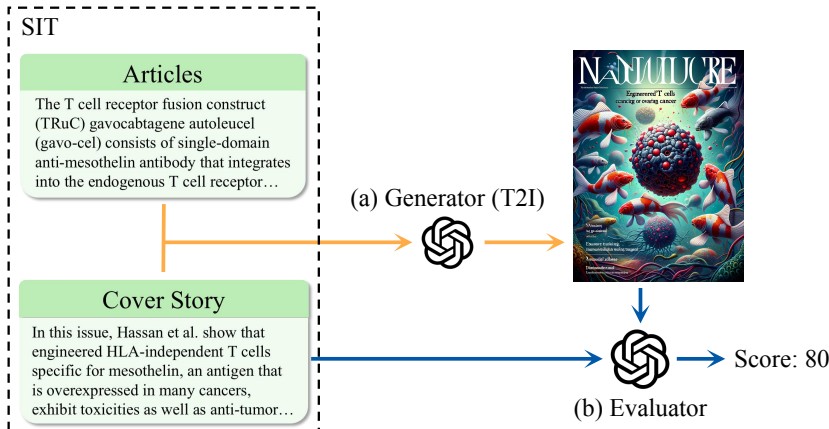

Figure 4: **The Text2Image task is to generate a journal cover given the articles and the cover story.** Following the yellow arrow, we provide articles and cover stories to a Generator, composed of either LMMs or our proposed Multimodal Agent Linkage. The Generator creates a cover image. Furthermore, as indicated by the blue arrow, this cover image is scored by our Evaluators, who are provided with the original cover story for reference.

concepts. Besides, such data is less probable to have been encompassed within the training datasets, thereby presenting a more formidable test for LMMs. So, based on the evolving nature of MAC, the most recent covers and cover stories have been chosen to construct the MAC-Latest dataset (50 issues) and MAC-Recent dataset (940 issues). These selections also make our evaluation fast and convenient. To ensure representativeness, efforts have been made to include covers and cover stories from a variety of journals, thereby encompassing a wide range of academic fields.

## 3.3 TASK SETUP

After obtaining covers, cover stories, and articles from a substantial number of issues, we have devised two tasks to assess different dimensions of the targeted multimodality and abilities of LMMs.

**Image2Text: (Cover, Article) to Cover Story**    As shown in Figure 3, we present the LMMs with a cover from an issue and the scientific facts extracted from the articles and task them with generating a formal cover story. Successful completion of this task necessitates the comprehension of LMMs for the implied artistic meaning and scientific manifestations depicted in the visual elements instead of relying on superficial concepts.

**Text2Image: (Cover Story, Article) to Cover**    As shown in Figure 4, LMMs are tasked with generating cover images based on provided cover stories and scientific articles. The goal is to assess their ability to visualize complex and elusive scientific phenomena or entities, as well as their capacity to understand and creatively represent advanced scientific concepts in a visually engaging manner.

## 4 METHOD

In this section, we discuss the Multimodal Agent Linkage (MAL) in Section 4.1. Furthermore, we demonstrate the combination of MAL and in-context learning techniques in Section 4.2.

## 4.1 MULTIMODAL AGENT LINKAGE

MAL is proposed to strengthen the ability of LMMs in the long-context window to comprehend and generate academic journal covers. LMMs such as MidJourney (Midjourney, 2023) and Stable-Diffusion (Rombach et al., 2021) excel in generating daily objects but struggle with understanding long-context language descriptions. Instead of finetuning LMMs for the downstream tasks, we view LMMs as agents and integrate them with Large Language Models (*e.g.*, ChatGPT (OpenAI, 2023),

ChatGLM (Zhipu, 2023)), which excel in long-context processing and understanding scientific concepts. This approach is simple and convenient, offering plug-and-play functionality. The multimodal integration mechanism can facilitate the exchange of information between lengthy contexts and abstract imagery. Additionally, MAL can leverage the unique strengths of each agent, enabling superior performance when tackling more challenging problems, which will be discussed in Section 5.3.

In the Image2Text task, Large Language Models (LLMs) leverage their generative capabilities to enhance the expressiveness of LMMs. Specifically, the role of LLMs is to refine and formalize the descriptions generated by MidJourney (Midjourney, 2023). This involves transforming the less structured outputs from MidJourney (Midjourney, 2023) into polished and structured cover stories suitable for professional and academic purposes. In the Text2Image task, LLMs use their ability to identify key points to grasp essential parts of lengthy texts and help LMMs understand abstract concepts. Specifically, LLMs are used to translate formal and content-focused cover stories into prompts that are stylistically aligned with the format requirements of MidJourney (Midjourney, 2023) and Stable-Diffusion (Rombach et al., 2021) during usage.

## 4.2    MAL with In-context Learning

In-context learning techniques are introduced to facilitate the cooperation between LMMs and LLMs in our proposed MAL when facing long-context language descriptions. Our initial experiments reveal that the direct outputs (0-shot) of MAL do not match academic journals in linguistic and artistic styles. Additionally, Large Language Models (LLMs) often alter the text to a huge degree from the images in Image2Text. Besides, summarizing cover stories by LLMs does not effectively help LMMs create covers in Text2Image tasks. Therefore, we integrate prompting techniques to combine their functionalities seamlessly. To investigate the differences among various prompting techniques, we include few-shot, chain-of-thought, and self-consistency prompting in our experiments for MidJourney and Stable-Diffusion. We find that these prompting methods fail in our cases and thus only display the few-shot results.

The details of four distinct prompting methods are as follows. In *zero-shot prompting*, LLMs are tasked with revising descriptions into cover stories and creating prompts for large multimodal models. *Few-shot prompting* (Brown et al., 2020) involves presenting three examples to guide LLMs in revision or prompt generation. In *chain-of-thought prompting* (Wei et al., 2022), a single example is given, and LLMs construct a chain of thoughts to guide their revisions or prompt generation. Lastly, *self-consistency prompting* (Wang et al., 2022) requires LLMs to produce three iterations of revised cover stories or prompts, with the most consistent one selected as the final output.

## 5    Experiment

Experimental settings are detailed in this section, including evaluated LMMs (Section 5.1) and the multi-faceted evaluation mechanism (Section 5.2). Ultimately, we analyzed the experimental results and the analysis (Section 5.3).

## 5.1    Large Multimodal Models

Gemini (Anil et al., 2023), MiniGPT-4 (Zhu et al., 2023), LLaMA-Adapter (Zhang et al., 2023) and LLaVA (Liu et al., 2023c) are LMMs that utilize image inputs to generate corresponding textual narratives. These models are applied in the Image2Text task for the creation of formal cover stories, drawing from provided journal covers and abstracts. For the widely perceived more capable Gemini (Anil et al., 2023), we specifically conducted experiments in MAC-Latest, to test its limits with more difficult issues.

CogView (Ding et al., 2021), GLM-4V (Zhipu, 2023), DALL·E 3 (OpenAI, 2023a) and GPT-4V (OpenAI, 2023b) are cutting-edge AI models designed for generating images and texts. Image2Text involves the curation of a varied collection of journal covers, upon which GPT-4V and GLM-4V are tasked to generate appropriate academic cover stories. In Text2Image, DALL·E 3 and CogView utilize these cover stories and abstracts to create artistically appealing and factually accurate covers. We evaluate DALL·E 3 and GPT-4V in the most challenging split of our benchmark, MAC-Latest, to reflect their true ability. GPT-4V is evaluated in MAC-Latest encompassing a carefully selected,

Table 2: **Automated evaluation preference is close to human experts.** In Image2Text on MAC-Latest, ChatGPT OpenAI (2023) is employed in the T-T assessment. Human experts are provided the original covers for a scoring reference(T-I). The currently recognized best, GPT-4V OpenAI (2023b), serves as the baseline for comparison. The scores are the average gap between the models and GPT-4V, calculated as avg_score($x$) − avg_score(GPT-4V).

| LLM | ChatGPT | | Human Experts | |
|---|---|---|---|---|
| | T-T Score | Avg. Ranking ↓ | T-I score | Avg. Ranking ↓ |
| GPT-4V | 0.0 | 2 | 0.0 | 2 |
| Gemini | (-2.0) | 3 | (-1.8) | 3 |
| MAL (MJ+GLM) | (+2.5) | 1 | (+4.0) | 1 |

Table 3: **LMMs' performance on MAC is generally poor.** The above are the results of Text2Image tasks evaluated by LLaVA and the below are the results of Image2Text tasks evaluated by GLM-4(T-T). Due to the demand of difficulty and cost constraints, DALL·E 3 was tested on a more challenging dataset. The ranking is based on the performance on MAC-Latest.

| LMM | MAC-Latest | | MAC-Recent | | MAC | | Ranking ↓ |
|---|---|---|---|---|---|---|---|
| | Score | Δ Score | Score | Δ Score | Score | Δ Score | |
| CogView-3 Ding et al. (2021) | 82.6 | (+0.0) | 82.6 | (+0.0) | 82.8 | (+0.0) | 2 |
| DALL·E 3 OpenAI (2023a) | 83.1 | (+0.5) | 83.3 | (+0.7) | 83.5 | (+0.7) | 1 |
| Stable-Diffusion Rombach et al. (2021) | 80.1 | (-2.5) | 81.2 | (-1.4) | 81.5 | (-1.3) | 3 |
| LLaVA Liu et al. (2023c) | 82.9 | (+0.0) | 83.5 | (+0.0) | 84.0 | (+0.0) | 3 |
| GLM-4V ZHIPU (2024) | 65.3 | (-17.6) | 65.3 | (-18.2) | 67.7 | (-16.3) | 6 |
| MiniGPT Zhu et al. (2023) | 84.8 | (+1.9) | 85.0 | (+1.5) | 85.1 | (+1.1) | 1 |
| LLaMA-Adapter Zhang et al. (2023) | 81.1 | (-1.8) | 81.9 | (-1,6) | 82.3 | (-1.7) | 5 |
| GPT-4V OpenAI (2023b) | 84.5 | (+1.6) | 84.8 | (+1.3) | 85.2 | (+1.2) | 2 |
| Gemini Anil et al. (2023) | 82.3 | (-0.6) | 82.4 | (-1.1) | 82.7 | (-1.3) | 4 |

diverse subset of the 50 most recent journal issues. DALL·E 3 is assessed on a larger collection of 940 recent issues, MAC-Recent.

## 5.2 EVALUATION

### 5.2.1 HUMAN EXPERT EVALUATION

History and collective memory have formed our universal values, creating a basis for judgment. When we solicit assessments from human experts essentially, we are anchoring their scoring to an ideal archetype ingrained within their consciousness. Specifically, in the Image2Text task, for MAC-Latest, we employ a human evaluation method. Four human experts are presented with the original cover alongside cover stories generated by GPT-4V, Gemini, and MAL. They assign scores by referencing the original cover for the generated cover stories, focusing on the literary and scientific merits of the generated content, and its reflection of the cover research's depth and context. Similarly, in the Text2Image task, DALL·E 3's performance on MAC-Recent was assessed through human evaluation to determine the alignment of AI-generated images with human preferences. A specialized website has been developed, offering clear guidelines and examples to assist field experts in their evaluations, which will be released later.

### 5.2.2 AUTOMATED EVALUATION

For the Image2Text task, we evaluate LMMs in two manners, *T-T Score* and *T-I Score*. *T-T Scores* involves a comparative analysis between the generated and original cover stories. Evaluation agents like ChatGPT, GLM-4, and GPT-4 are supplied with two versions of cover stories and assign scores based on artistic and academic values. *T-I Scores* examine the correlation between the generated cover stories and the original journal covers. The journal covers and the generated cover stories in MAC-Latest are provided to the evaluation agent, tasked with discerning their interrelation. To

Table 4: **MJ+GPT's performance on MAC across academic disciplines evaluated by GPT-3.5 and T2I performance evaluated by LLaVA.**

| Category | # Issues | Avg. I2T Score | Avg. T2I Score |
|---|---|---|---|
| Biology | 3693 | 77.2 | 82.6 |
| Chemistry | 155 | 78.2 | 82.6 |
| Geology | 83 | 77.5 | 82.9 |
| Engineering | 292 | 79.1 | 81.8 |
| Ecology | 264 | 77.6 | 82.1 |
| Materials Science | 93 | 79.1 | 81.8 |
| Oncology | 423 | 77.7 | 82.2 |
| Miscellaneous | 374 | 78.2 | 82.1 |
| Astronomy | 441 | 78.7 | 82.7 |
| Sociology | 54 | 75.2 | 80.5 |

enhance the accuracy of this assessment, two GPT-4V agents are deployed, each focusing on artistic and scientific comprehension, respectively.

For the Text2Image task, the metric *I-T Score* employs hinges on the similarity between former cover stories and the generated covers. Evaluation agents give scores based on both scientific and artistic dimensions. In the GPT-4V evaluation of MAC-Latest, scores reflecting scientific and artistic understanding are output separately. For MAC, LLaVA is used as an alternative for assessing the entire benchmark.

Automated evaluation shows a highly similar preference to human experts. Table 2 compares the performance of GPT-4V, Gemini, and our approach MAL (MJ+GLM) with automatic evaluation and human expert evaluation. Automatic evaluators are shown with comprehensive and actionable criteria for evaluation. It has proven to be closely aligned with human preference and can accurately reflect the strengths and weaknesses of evaluated models.

## 5.3 EXPERIMENTAL RESULT

Most existing models perform poorly on both Text2Image and Image2Text tasks. Table 3 shows the rankings and scores of existing LMMs. Their performance is still far from satisfactory. An important trend emerges in Table 3: the performance of models deteriorates as they face more challenging subsets. This outcome suggests that LMMs heavily rely on recalling their trained memory. However, when confronted with untrained, unknown knowledge, they fail to utilize existing knowledge for reasoning and constructing a knowledge framework.

Disciplines with higher frequency in our MAC seem to perform better in current LMMs. Table 4 shows the performance of MJ+LLM on Image2Text and Text2Image tasks. We classify the dataset based on Web of Science [4]. We find that LMMs perform the worst in Sociology, which has the least issues in all journals. Social sciences pose greater challenges in representation, unlike other disciplines that are based on physical phenomena and easier to conceptualize. Besides, we find a rough trend where LMMs perform better in disciplines with more issues in all journals. We reckon that the mainstream disciplines are abundant with training data for LMMs, thus showing the best performance.

Modular MAL is adept at handling complex issues. According to Table 5 , MAL shows a significant improvement in handling the Text2Image and Image2Text tasks compared to the bare LMMs when facing long-context language windows. The information exchange between two modalities links the two agents together, allowing each to leverage its respective strengths. Table 3 compares the performance of DALL·E 3 (OpenAI, 2023a), Stable-Diffusion (Rombach et al., 2021) (SD), and CogView (Ding et al., 2021) across various dataset scales. It's interesting to note that when the leading DALL·E 3 shows a decline in performance in MAC-Latest, MAL like ChatGLM+SD exhibits improved performance, with consistent improvements or minimal declines, according to Table 5.

---

[4]https://incites.help.clarivate.com/Content/Research-Areas/wos-research-areas.htm

Table 5: **Our proposed MAL has exhibited enhanced efficacy in both Text2Image (top) and Image2Text (bottom) tasks.** For Text2Image tasks, these experiments are conducted across various scales of MAC and are subjected to evaluation by the LLaVA Liu et al. (2023c) framework. For Image2Text tasks, we introduce GLM-4 ZHIPU (2024) as the evaluation agents.

| Method | MAC | | MAC-Recent | | MAC-Latest | |
|---|---|---|---|---|---|---|
| | Score | Δ Score | Score | Δ Score | Score | Δ Score |
| Stable-Diffusion Rombach et al. (2021) | 81.5 | (+0.0) | 81.2 | (+0.0) | 80.1 | (+0.0) |
| + MAL-GLM ZHIPU (2024) | 82.3 | (+0.8) | 82.3 | (+1.1) | 82.1 | (+2.0) |
| MidJourney Midjourney (2023) | 48.1 | (+0.0) | 47.5 | (+0.0) | 47.3 | (+0.0) |
| + MAL-GLM ZHIPU (2024) | 77.4 | (+29.3) | 79.8 | (+32.3) | 80.2 | (+32.9) |

Table 6: **In Image2Text, the MAL (MJ+LLM) demonstrates varied capabilities when applied with different prompt techniques.** The whole MAC is tested using the MidJourney+ChatGLM and MidJourney+ChatGPT with the evaluators of GPT-3.5 and GPT-4.

| Model | Method | T-T (GPT-3.5) | T-T (GPT-4) |
|---|---|---|---|
| ChatGLM Zhipu (2023) | 0-Shot | 81.0 | 52.1 |
| | 3-Shot | 75.9 | 51.7 |
| ChatGPT OpenAI (2023) | 0-Shot | 75.6 | 65.1 |
| | 3-Shot | 77.6 | 65.1 |

Advanced prompting may result in poorer performance due to model limitations. Table 7 provides a comparison of different in-context learning techniques in MAL. Comparing ChatGLM and ChatGPT using 0-shot and 3-shot approaches, we find that in-context learning techniques do not necessarily yield better results. In-context learning imposes certain requirements on the model's ability to comprehend long texts, which can harm models that are not proficient in understanding lengthy passages. We also experiment with the chain-of-thought and self-consistency prompting but find these prompting methods fail in our cases.

DALL·E 3 (OpenAI, 2023a) still needs improvement in understanding context-specific terms, producing diverse details, and accurately capturing text. DALL·E 3 exhibits factual errors in depicting objective objects especially related to culture, as it fails to capture some of their typical characteristics: inaccuracies in depicting *Monkey King* and *White Bone Demon*. It also struggles to understand idioms within context: *Two Tigers cannot Hide in the Same Mountain* expresses the existence of opposition and conflict, which cannot be represented by simply drawing a tiger.

## 6 CONCLUSION

This paper introduces the Multimodal Academic Cover (MAC) benchmark, a novel evaluation framework derived from prominent academic journals such as Cell, Nature, and Science. Our focus lies on highly abstract and conceptual content. We craft bidirectional generative tasks centered around a single journal issue, fostering a nuanced comprehension of academic covers from both perspectives. Through our evaluation, we delve into the performance of advanced LMMs, revealing their shortcomings in handling these tasks effectively. To address these limitations and enhance the capabilities of LMMs, we propose Multimodal Agent Linkage (MAL), a collaborative framework wherein LMMs and LLMs synergize to leverage their strengths. Through MAL, we demonstrate a significant improvement in the understanding of scientific concepts, particularly in tackling more complex tasks. Moreover, we underscore the significance of multi-directional generative benchmarks and outline our intention to explore cyclic generation techniques in future research, aiming for a more holistic understanding of LMMs.

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

# A  ADDITIONAL EXPERIMENT RESULTS

The supplementary materials are provided below. Firstly, chain-of-thought (CoT) (Wei et al., 2022) and self-consistency (SC) (Wang et al., 2022) prompting results are shown in Appendix A.1 to indicate their inability to enhance Multimodal Agent Linkage (MAL)'s performance. Secondly, in Appendix A.2, the detailed analysis of the generation of Large Language Models (LMMs) and MAL emphasize MAL's advantages. Lastly, the comprehensive and detailed evaluation criteria are demonstrated in Appendix A.3.

## A.1  PROMPTING DOES NOT NECESSARILY IMPROVE MAL.

According to Table 7, when applying CoT (Wei et al., 2022) and SC (Wang et al., 2022) promptings, both MAL (MJ (Midjourney, 2023)+ChatGLM (Zhipu, 2023)) and MAL (MJ+ChatGPT (OpenAI, 2023)) show unsatisfactory performance compared to the 0-Shot prompting. Although CoT prompting provides more logical reasoning which enhances the ability in the scientific dimension, it does little to the artistic dimension, thus leading to a weak ability in cover generation. Moreover, ChatGLM and ChatGPT's generations lack diversity so SC prompting can't identify the optimal one.

Table 7: **MAL demonstrates poor capabilities when applied with advanced prompt techniques in Image2Text experiments conducted on MAC.** The whole MAC is tested on the MAL (MJ+ChatGLM) and MAL (MJ+ChatGPT) with the evaluators of ChatGPT.

| Model | Method | T-T (ChatGPT) |
|---|---|---|
| ChatGLM | 0-Shot | 81.0 |
| | 3-Shot (Brown et al., 2020) | 75.9 |
| | CoT (Wei et al., 2022) | 76.6 |
| | SC (Wang et al., 2022) | 78.6 |
| ChatGPT | 0-Shot | 75.6 |
| | 3-Shot (Brown et al., 2020) | 77.6 |
| | CoT (Wei et al., 2022) | 74.4 |
| | SC (Wang et al., 2022) | 74.7 |

## A.2  MAL AND LMMS DEMONSTRATE DISTINCT STYLES IN BIDIRECTIONAL TASKS.

The detailed performance of our tested LMMs and MAL are shown. For Text2Image tasks, the second image in Figure 5 depicts the 3D spatial relation between the leopard and a tree. But DALL·E 3 (OpenAI, 2023a) shows an evident cartoon style. In contrast, according to Figure 6, MAL masters various painting skills including the abstract (1st), cartoon (2nd), and realistic (6th) styles. Moreover, the 8th generated cover depicts the speed of the cars, which reflects MAL's understanding of physics laws.

For Image2Text tasks, the comparisons in Figure 7, Figure 9, and Figure 8 showcase LMMs' capabilities in composing cover stories, particularly highlighting their proficiency in artistic interpretation, literary expression, and adherence to scientific facts. Interestingly, cover stories generated by GPT-4V (OpenAI, 2023b) and Gemini (Anil et al., 2023) tend to be redundant. On the contrary, MAL's generations are more concise. MAL proficiently begins its cover story with attractive words, such as "welcome to a realm where the fusions ...". Besides, MAL establishes evident logical connections between sentences.

## A.3  THE CRITERIA FOR BOTH HUMAN-INVOLVED AND AUTOMATIC EVALUATIONS ARE COMPREHENSIVE AND DETAILED.

For more valid evaluations, we meticulously crafted criteria for both manual and automatic evaluation. Figure 10 displays the standards referred to in human evaluations, which are available on our evaluation website. Human experts will score based on our established criteria, ensuring consistency

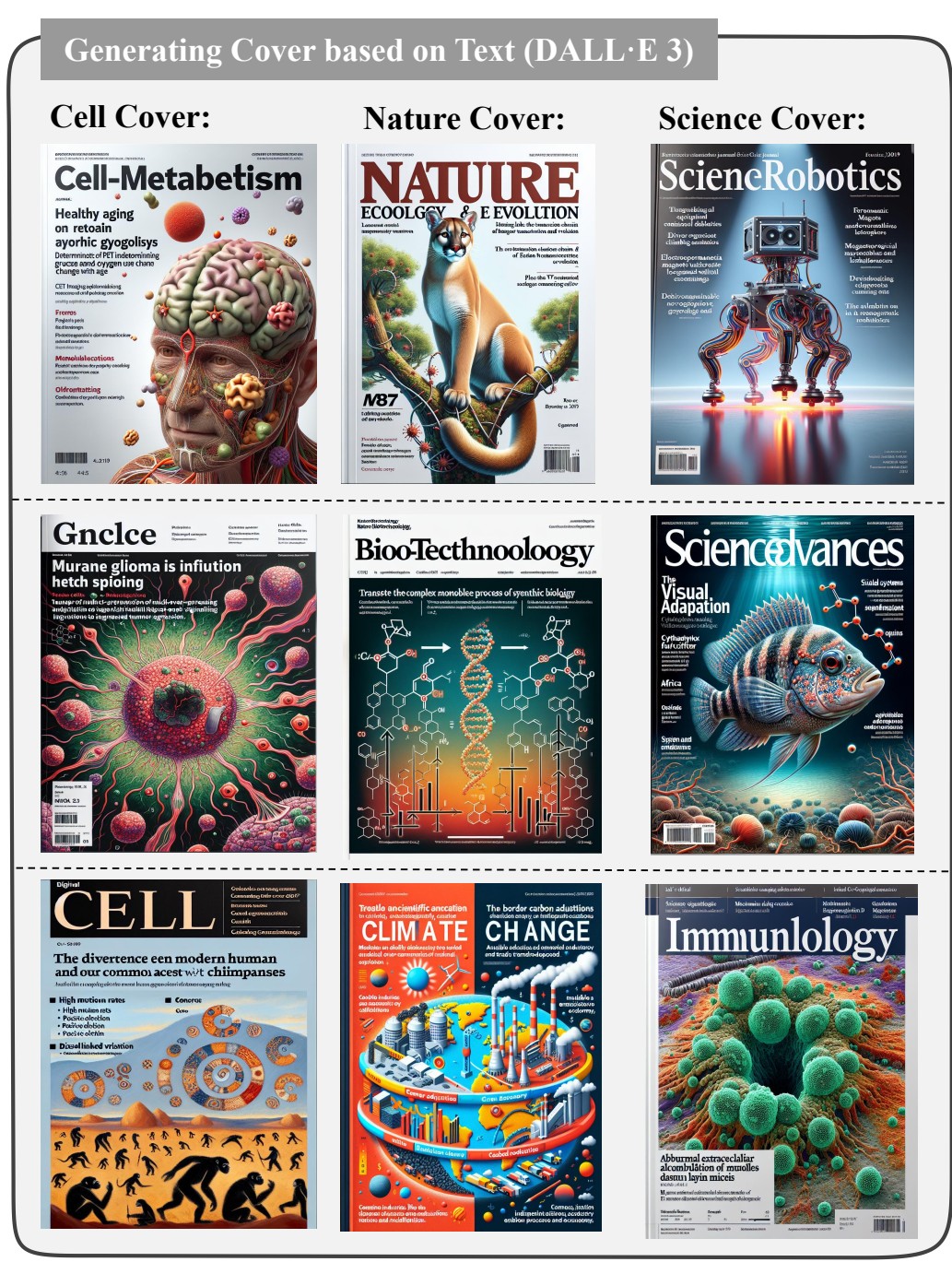

Figure 5: **The Text2Image performance of DALL·E 3 (OpenAI, 2023a) on the three journal families is shown above.** DALL·E 3 demonstrates amazing performance on cover generation, especially the journal cover design. Images are labeled from left to right follows: 1st, 2nd, and 3rd (top row); 4th, 5th, and 6th (mid row); 7th, 8th, and 9th (bottom row).

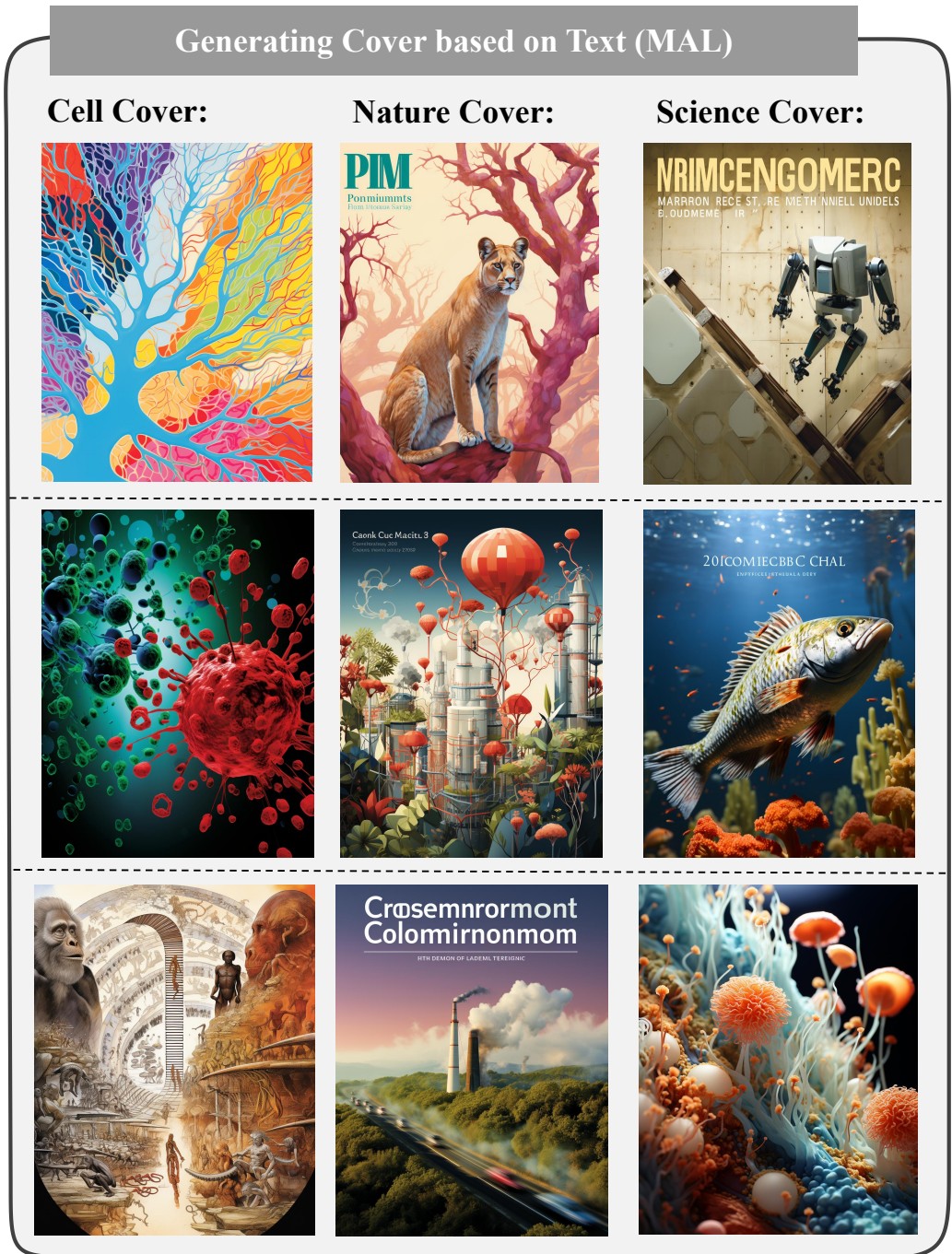

Figure 6: **The Text2Image performance of MAL (GLM+MJ) on the three journal families is shown above.** MidJourney (MJ) (Midjourney, 2023) is combined with ChatGLM (Zhipu, 2023) to construct a MAL using CoT prompting (Wei et al., 2022). The shown issues are the same as those DALL·E 3 (OpenAI, 2023a) operates on for a comparison. Images are labeled from left to right follows: 1st, 2nd, and 3rd (top row); 4th, 5th, and 6th (mid row); 7th, 8th, and 9th (bottom row).

## Generating Text based on Cover (GPT-4V)

**Cell :**

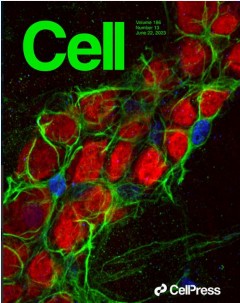

At the frontier of biomedical research, we unravel the confounding enigma of how psychological stress seeds havoc in the labyrinth of the human body. In a seminal exploration, Schneider et al. illuminate the previously obscure pathways through which stress hormones sculpt the landscape of our enteric nervous system, establishing a direct correlation between the storms of the mind and the fires of gut inflammation. The vibrant interplay of fluorescent hues on display is less an artistic masterpiece and more a map of uncharted physiological territory, guiding us through the interactions of glucocorticoids with the cellular sentinels of our gut. As the colors blend, so too does our understanding deepen, of how the invisible tendrils of stress tighten their grip on the most primal aspects of our health. This groundbreaking research not only peels back the layers of complex biologic interactions but also sews the seeds for novel therapeutic horizons, promising alleviation for those afflicted by inflammation's relentless march.

**Nature :**

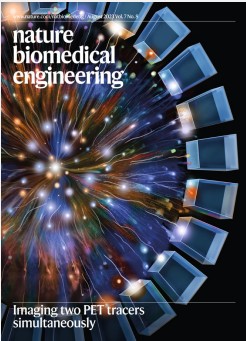

In an unprecedented fusion of scientific acumen and technological prowess, we capture the essence of innovation with the groundbreaking advancement in biomedical imaging: multiplexed positron emission tomography (PET). This radical approach shatters the status quo, harnessing the full spectrum of annihilation photons and prompt gamma-ray emissions, which were once elusive in simultaneous imaging endeavors. By defying traditional limitations, investigators have now elucidated the distinct pathways of two PET tracers in vivo, endowing researchers with the uncanny ability to dissect complex molecular dialogues within living organisms. The multiplexed PET technique celebrates the union of precision and efficiency, reducing patients' exposure to radiation and circumventing the need for ancillary diagnostic procedures. The scintillating visual on our cover captures the very spirit of this breakthrough: a convergence of light and energy, symbolizing the newfound clarity in visualizing concurrent biological events, and thus, opening vistas in diagnostic imaging that were once veiled in shadow.

**Science :**

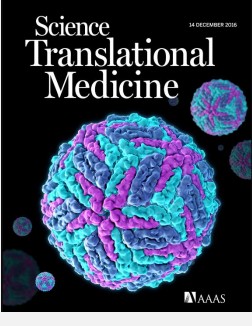

In the wake of the devastating impact wrought by the 2015-2016 outbreak of Zika virus, the pursuit of therapies to combat the emerging threat has intensified. Amidst this landscape of urgency, a beacon of hope emerges as the isolation of specific human monoclonal antibodies presents a novel avenue for intervention. These antibodies, identified from a patient with previous infection, offer formidable ZIKV neutralization capabilities while sparing the closely related dengue virus strains. Key to this strategy is the profound understanding of the interactive sites on the virus envelope proteins—domain I, II, or III—where these antibodies exhibit their potent activity, insights that illuminate a path forward for therapeutic and vaccine design. The synergy between structural biology and translational medicine is exquisitely captured in the findings reported, promising a dedicated effort in battling a public health nemesis and providing a cornerstone in the edifice of infectious disease control.

Figure 7: **The Image2Text performance of GPT-4V (OpenAI, 2023b) on the three journal families is shown above.** Three representative cover stories generated by GPT-4V were chosen. The left side displays the original cover images, while the right side showcases the corresponding cover stories. We provide a comparison of the performance across different journals, highlighting the interpretation of artistic elements and the expression of scientific facts .

## Generating Text based on Cover (Gemini)

**Cell :**

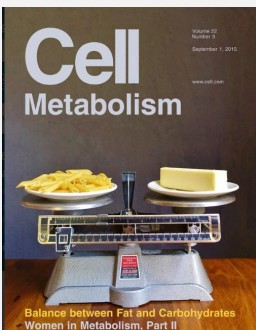

The cover of this issue of Cell Metabolism features a simple yet powerful image: a plate of pasta and a plate of butter on a scale. The pasta plate is slightly heavier than the butter plate, suggesting that pasta contains more energy per gram than butter. This image is a visual representation of the scientific breakthrough reported in the cover article: cutting fat from your diet may be more effective for weight loss than reducing carbohydrates, even when the two diets are matched for calories. This finding challenges the conventional wisdom that low-carbohydrate diets are the most effective way to lose weight. It also raises important questions about the mechanisms by which different macronutrients affect weight loss. The study's findings are based on a small group of participants, so more research is needed to confirm the results in larger and more diverse populations. However, the study provides a strong starting point for future research on the role of fat and carbohydrates in weight loss.

**Nature :**

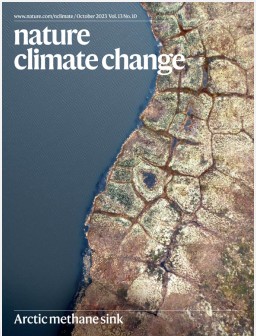

The Arctic is a region in crisis. Climate change is causing the permafrost to thaw, releasing methane, a potent greenhouse gas, into the atmosphere. This is a vicious cycle, as methane further warms the planet, causing more permafrost to thaw. However, there may be a new hope for the Arctic. A recent study published in _Nature Climate Change_ has shown that well-drained Arctic soils can consume atmospheric methane at significant rates. This suggests that Arctic soils may play a more important role in the global methane cycle than previously thought, and could even provide a negative feedback to global climate change. This study is significant because it offers a potential solution to the problem of methane emissions from the Arctic. If we can find ways to promote methane uptake by Arctic soils, we could help to slow the warming of the Arctic and mitigate climate change. More research is needed to understand the full potential of Arctic soils as a methane sink. However, the findings of this study are a promising sign that the Arctic may not be doomed after all.

**Science :**

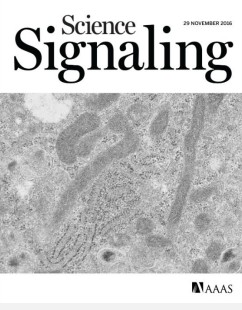

The cover of this issue of Science Signaling depicts a stylized image of a mitochondrion, the powerhouse of the cell. The mitochondrion is engulfed in flames, representing the oxidative stress that is characteristic of systemic lupus erythematosus (SLE). Within the mitochondrion, a protein called MAVS is oligomerizing, forming aggregates that trigger the production of type I interferons. These interferons are essential for fighting viral infections, but in SLE, they are produced in excess, leading to a cytokine storm that can damage tissues and organs. The cover image accurately captures the key finding of the cover article by Dr. Lauriola and colleagues: oxidative stress-induced MAVS oligomerization contributes to the type I interferon signature that is characteristic of SLE. This finding has important implications for the development of new treatments for SLE, as it suggests that targeting MAVS oligomerization or its downstream signaling pathways could be an effective way to reduce inflammation and improve patient outcomes.

Figure 8: **The Image2Text performance of Gemini (Anil et al., 2023) on the three journal families is shown above.** Three representative cover stories generated by Gemini were chosen. The left side displays the original cover images, while the right side showcases the corresponding cover stories. We provide a comparison of the performance across different journals, highlighting the interpretation of artistic elements and the expression of research facts.

## Generating Text based on Cover (MAL)

### Cell :

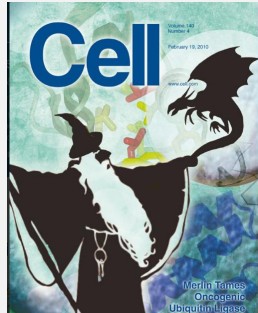

Welcome to a realm where the fusion of art and science transcends boundaries, as our cover mesmerizes with the intricate dance between a wizard and a dragon, inspired by the artistic brilliance of Fernando Amorsolo, Tatsuyuki Tanaka, and Amedeo Modigliani, presented in the style of molecular intricacy. In this issue, the enigmatic FERM domain protein Merlin, guardian of the NF2 tumor suppressor, unravels a compelling narrative. Its unexpected translocation to the nucleus, binding to the E3 ubiquitin ligase CRL4DCAF1, and subsequent tumor-suppressive program of gene expression redefine our understanding of tumorigenesis. Join us on a journey where scientific inquiry and artistic inspiration converge to illuminate the complex tapestry of cellular intricacies, forging new pathways towards transformative discoveries.

### Nature :

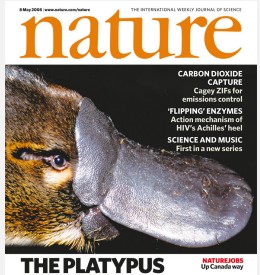

In this edition, we delve into the genetic wonders of the platypus, a true masterpiece of nature's artistry. Its genetic composition, akin to a symphony of "hatecore" and organic fluidity, reveals the secrets behind its reptilian and mammalian fusion. With distinctive noses as their signature, platypuses challenge our understanding of evolution and adaptation. The platypus genome's revelations, from venomous co-options to conserved milk proteins and immune gene expansions, mark a turning point in genetic exploration. This genomic treasure trove opens doors to profound insights into mammalian evolution, making it an essential tool for future comparative studies. As we delve into the fluidity of platypus genetics, we embark on a journey to conserve and protect this unique species. Join us in celebrating the essence of science and the wonders of nature.

### Science :

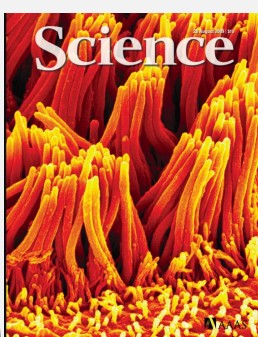

In the vivid realm of cellular life, this edition's cover art hints at the awe-inspiring revelations within. Cilia, those intricate projections emerging from eukaryotic cells, have long been a subject of fascination in the scientific world. In this issue, we explore the astonishing capabilities of motile cilia within human airway epithelial cells. Their dual role as defenders and sentinels is a testament to the intricate design of nature. Sensory bitter taste receptors on motile cilia awaken a cellular response, unveiling a remarkable system that senses and combats harmful intruders, echoing the sensory prowess of primary cilia. Science, as showcased in these pages, never ceases to amaze, even in the microscopic realm.

Figure 9: **The Image2Text performance of MAL (MJ+GLM) on the three journal families is shown above.** Three representative cover stories generated by MAL (ChatGLM (Zhipu, 2023)+Mid-Journey (Midjourney, 2023)) with SC prompting (Wang et al., 2022) were chosen. The left side displays the original cover images, while the right side showcases the corresponding cover stories. We provide a comparison of the performance across different journals, highlighting the interpretation of artistic elements and the expression of research facts.

of scores. In the automatic evaluation, evaluators are required to provide their reasoning for their scores, which is interpretable. Two agents are introduced for detailed and comprehensive measurement of artistic and scientific aspects. Figure 11 and Figure 12 illustrate examples of two agents' rationale for two distinct tasks, which strictly adhere to the standards and perspectives provided in our prompts.

**Instruction**: You will receive an image and accompanying text from authoritative scientific journals like Science, Cell, and Nature. The text, written by the magazine's editor-in-chief, introduces the cover illustration and its visual elements. It explains the artist's design choices and highlights the scientific value of the cover article.

Your task is to evaluate the cover design's quality based on the given text and assign a score from 0 to 100. Consider both aesthetic value and scientific understanding:

**[Aesthetic Value]**
**Composition and Design**: Is the cover well-structured and visually pleasing?
**Color Harmony**: Do the colors used in the cover harmonize and match the described mood or tone?
**Atmosphere**: Does the cover effectively evoke the intended atmosphere or mood?
**Creativity**: Does the cover demonstrate artistic ingenuity and unique elements that capture attention?

**[Scientific Understanding]**
**Text-Image Relevance**: Does the image strongly relate to the content described in the text? Can a reader infer the research direction from the image?
**Scientific Accuracy**: Does the depiction of scientific concepts on the cover reasonably and accurately represent factual information?
The reference baseline score is 75. We appreciate your contribution and value your evaluation. Here are two examples for you:

Score: 70
Reason: The image is visually impactful and creative which contrasts the different conditions of two types of trees in a fire but does not further illustrate the reasons.

Score: 80
Reason: The image reflects the drawing inspiration mentioned in the editorial and embodies the process of pre-invasive lesion. It is highly expressive.

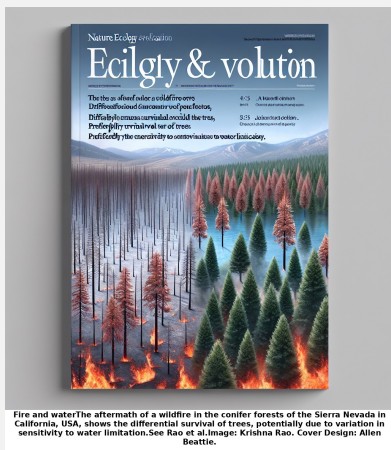

Fire and water.The aftermath of a wildfire in the conifer forests of the Sierra Nevada in California, USA, shows the differential survival of trees, potentially due to variation in sensitivity to water limitation.See Rao et al.Image: Krishna Rao. Cover Design: Allen Beattie.

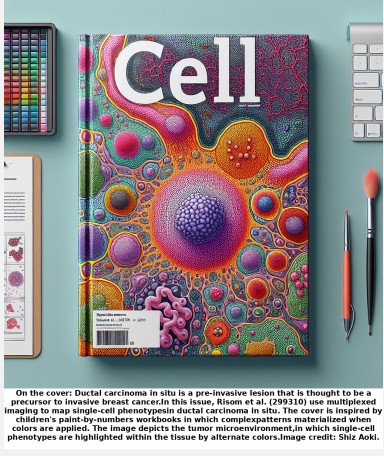

On the cover: Ductal carcinoma in situ is a pre-invasive lesion that is thought to be a precursor to invasive breast cancer.In this issue, Risom et al. (299310) use multiplexed imaging to map single-cell phenotypesin ductal carcinoma in situ. The cover is inspired by children's paint-by-numbers workbooks in which complexpatterns materialized when colors are applied. The image depicts the tumor microenvironment,in which single-cell phenotypes are highlighted within the tissue by alternate colors.Image credit: Shiz Aoki.

Figure 10: **In the context of the Text2Image task for DALL·E 3 (OpenAI, 2023a), we aim to assess the quality of AI-generated scientific covers accurately.** To achieve this, we introduce a comprehensive human evaluation process conducted on a large-scale test dataset. The provided instructions serve as guidance for the scorers involved in the evaluation process.

**The reasoning of GPT-4V scoring on Image2Text**

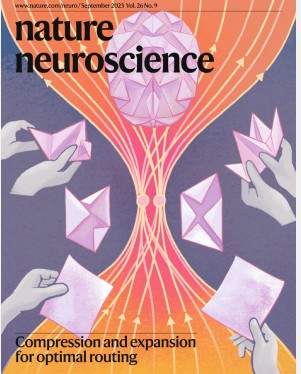

### Generated Cover Story

In an era where information exchange is pivotal, the power of crowd-sourced intelligence has crystallized in the form of OpenPath, a transformative dataset comprising over 200,000 pathology images intricately annotated with natural language descriptions. This edition showcases the bridging of computational sciences and clinical expertise, a synergy encapsulated by the cover art which artistically represents the emergence of pathology language-image pretraining (PLIP) through a vibrant mosaic of micrographs flowing into the digital ether. PLIP, a beacon in the frontier of medical artificial intelligence, heralds a quantum leap with its stellar zero-shot classification performances, eclipsing previous models and carving pathways for intuitive retrieval of case similarities by image or descriptive search. ......

**Art Bot: [92]**
The literary quality of the editorial is impressive based on the criteria provided:
- Fluency of language: The text is well-composed, with a smooth flow that makes it easy to follow. The use of advanced vocabulary and complex sentence structures does not hinder understanding but rather emphasizes the sophistication of the subject matter.
- Vividness of expression: ......
- Artistic understanding: ......
- Interest in content: ......

**Sci Bot: [95]**
The editorial shows a strong scientific quality :
- Relevance of content: The editorial is highly relevant to the cover image. The cover art features what appears to be pathology images forming a cloud-like formation infusing into a digital landscape, which symbolizes the intersection of pathology and artificial intelligence. The editorial correctly identifies this bridge between computational sciences and clinical expertise.
- Correct understanding: ......
- Specificity of content:......

### Generated Cover Story

We unveil the intricate ballet of neural architecture, where the cerebellar granule cells emerge as the prima ballerinas of synaptic grace. Guided by the pioneering insights on mossy fibers, a stunning realization unfolds that their dance with cerebellar granule cells is more than mere connectivity—it's a nuanced choreography enabling sophisticated computations necessary for associative and internal model learning. The graphical ensemble displayed radiates with vibrant hues and geometric constructs, a visual symphony that metaphorically captures the intense interactions of neuronal network. It mirrors the complexity and elegance within our brains, where a seemingly simple passage, a bottleneck, fosters a computational hub of information structuring, an essential prelude to the cerebellar expansion. ......

**Art Bot: [85]**
The literary quality of this editorial is high based on the criteria outlined.
- Fluency of language: The editorial is written with a high level of sophistication in its use of language. Phrases such as \"intricate ballet of neural architecture\" and \"vibrant hues and geometric constructs\" flow well and create a smooth narrative, although the complexity of the vocabulary and metaphor may challenge some readers. The text reads smoothly for the most part.
- Vividness of expression: ......
- Artistic understanding: ......

**Sci Bot: [92]**
The editorial does a commendable job in capturing the scientific essence depicted on the magazine cover of Nature Neuroscience, with a particular emphasis on the neural processes of compression and expansion within the cerebellum and their roles in neural routing and information processing. Here's the breakdown of the assessment:
- Relevance of content: ......
- Correct understanding: ......
- Specificity of content: ......

Figure 11: **The scores and reasons of Image2Text tasks GPT-4V (OpenAI, 2023b) are reasonable and thoughtful.** The original cover of the journal is shown above, while the accompanying cover story is generated using the GPT-4V model. The scoring process adheres to the predefined criteria provided in the prompts, ensuring that GPT-4V complies with our requirements and assigns scores based on these criteria rigorously.

## The reasoning of GPT-4V scoring on Text2Image

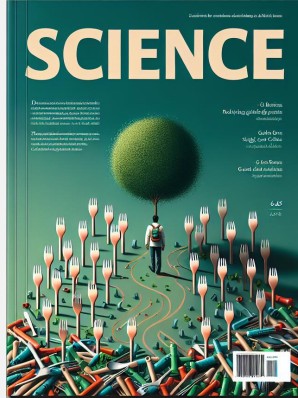

**Art Bot: [85]**
The magazine cover aligns with the editorial theme and executes an artistic depiction that is both appealing and appropriate to the described content.
-Composition and Layout: The cover features a well-arranged scene with a clear focal point: a person standing on trash piles of single-use cutlery, which leads up to a graph ......
- Color Harmony: The colors are harmonious and ......
- Medium and Style: There is a distinct ......
......

**Sci Bot: [90]**
- Overall Image-Text Alignment: The cover image is in strong alignment with the given editorial text. ......
- Conceptual Representation: The cover does an excellent job in portraying the abstract scientific concepts mentioned in the editorial.......
What slightly reduces the score is that the graph in the background is not directly referenced in the editorial description, and its relevance to the research findings is assumed rather than explicit. However, this does not significantly detract from the overall strong representation of the scientific editorial."

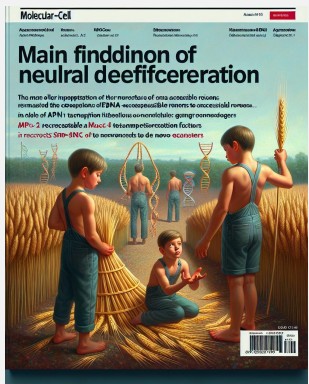

**Art Bot: [93]**
- Composition and Layout: The cover presents a central image of a mitochondrion that captures the viewer's attention, effectively becoming the focal point of the design.......
- Color Harmony: The colors used are harmonious and seem to be carefully chosen to reflect the biological subject matter......
- Medium and Style: ......
- Overall Aesthetics: Visually, the cover is striking and engaging. Despite the complexity of the scientific subject, the image is rendered in a way that is accessible and attractive to a broader ......

**Sci Bot: [95]**
The magazine cover appears to align very well with the editorial provided, with just minor points for improvement.
- Overall Image-Text Alignment: The cover image showcases a transmission electron micrograph of mitochondria which ties directly into the main subject of the editorial......
- Conceptual Representation: The cover explicitly shows the aggregation of MAVS in response to ......
Overall, the cover does an excellent job of capturing the essence of the research and presenting it in a manner that is both scientifically evocative and aesthetically engaging.

**Art Bot: [92]**
- Composition and Layout: The cover presents a well-composed scene......
- Color Harmony: The colors are vibrant and harmonious, blending warm and cool tones to create a visually pleasing image. The warm colors of the wheat and the fiery trees combine with......
- Medium and Style: ......
- Overall Aesthetics: ......
The cover effectively translates the complex scientific details into a visually engaging story, creating a successful and striking design that supports the editorial material in an innovative and artistic manner.

**Sci Bot: [85]**
Overall Image-Text Alignment:\nThe cover appears to accurately represent key elements from the given editorial, with some artistic license. The older boy with the sheaves of wheat could......
Conceptual Representation:The cover's conceptual representation is quite strong......
However, due to the abstract nature of biological processes, some precision is inherently lost in artistic translation, thus the score is not a full 100.

Figure 12: **The evaluation agent, GPT-4V (OpenAI, 2023b), gives scores and explanations of Text2Image tasks given.** The generated cover of the journal is shown above, which shows the ability of DALL·E 3 (OpenAI, 2023a). The scoring process adheres to the predefined criteria provided in the prompts, ensuring that GPT-4V complies with our requirements and assigns scores based on these criteria rigorously.

