# OpenReview forum: "MAC: A Multimodal Benchmark for Understanding and Generating Academic Journal Covers"
_ICLR.cc/2025/Conference — Submitted to ICLR 2025_

### Official Review · Reviewer_ekpe · 2024-10-27

**Soundness:** 3
**Presentation:** 3
**Contribution:** 2
**Rating:** 3
**Confidence:** 4

**Summary:**

The paper presents a benchmark dataset for the evaluation of text-to-image and image-to-text tasks. The dataset consists of close to 6000 cover images of scientific journals. The paper reported human-evaluation results of a number of well-known image-to-text and text-to-image models like DALLE3, GPT-4V, etc on this benchmark dataset.

**Strengths:**

The main contribution if the benchmark dataset. As far as I know, this is the first benchmark dataset on scientific journal cover image understanding/generation.

**Weaknesses:**

The main weakness is that the scope of the dataset is too narrow. It is limited to the cover images of three scientific journals (cell, science, nature). Conclusions drawn for the cover images in these journals may not be valid for many academic journals in other science and engineering disciplines.

 Since those journals are all well-known journals. The cover images may have been used by many of the models as training data. It is difficult to come up with new test images which have not been seen by the models.

Another problem is that there is no quantitative measure on what makes a good cover image. Some scientific journals have their cover pages simply listing the titles of all the papers or the titles of a few selected feature articles. Are those cover images considered good or bad?

**Questions:**

Is there any way to know if a model has already included the benchmark images into their training data?

Given an "imaginary" journal name, and an arbitrary article (with text and diagrams), there are many ways to come up with a "reasonable" cover image. Is there a quantitative way to measure the "quality" of the images?

How can the benchmark dataset be extended to other journals which may not show feature articles in their cover images?

---

### Official Review · Reviewer_WAv3 · 2024-10-28

**Soundness:** 2
**Presentation:** 2
**Contribution:** 2
**Rating:** 3
**Confidence:** 4

**Summary:**

This paper propose The Multimodal Academic Cover (MAC) benchmark to evaluate Large Multimodal Models (LMMs) in generating and understanding academic journal covers. MAC uses Image2Text and Text2Image tasks to assess current LMMs. Additionally, it introduces Multimodal Agent Linkage (MAL) to enhance conceptual comprehension within a long-context window.

**Strengths:**

This paper firstly introduces an benchmark for evaluating the LLM's ability on (1) generating cover for scientific journals and (2) understanding the cover of scientific journals

**Weaknesses:**

1. Benchmark lacks scalability and practicality. Can MAC extended to other types of covers?
2. Benchmark design details are relatively poor.
3. The proposed method Multimodal Agent Linkage (MAL) lacks innovation.
See questions below.

**Questions:**

1. I have doubts about the scalability and practicality of the MAC. Although the authors proposed a benchmark for the understanding and generation of scientific journal covers, I did not see the distinction between scientific journals and other types of documents in the context of this article's topic. For example, the generation and understanding of covers by LLMs for magazines, textbooks, etc., are almost identical to those of scientific journals, with no significant differences. From this perspective, why is the theme of the Benchmark limited to scientific journals? Or can MAC be extended to other types of covers?
2. The scoring mechanism of the MAC is too vague and lacks refinement. For instance, in the prompt of the appendix, only a few aspects such as color and relevance are mentioned. A more reasonable approach for LLMs would be to evaluate the model using different fine-grained metrics, then combine these scores using weighting methods to derive a final score.
3. I have concerns regarding the human-expert evaluation. For instance, in the I-T task, the reference stories provided should be highly specialized, requiring strong domain-specific knowledge to understand. The paper does not adequately consider or explain whether the human experts meet this criterion. LLMs might generate stories that contain similar terms and themes but could be logically incorrect or entirely wrong due to hallucination. Has this possibility been considered?
4. Why is the performance of GLM-4V worse than MiniGPT-4 in Table 3?

---

### Official Review · Reviewer_SQyG · 2024-11-02

**Soundness:** 2
**Presentation:** 3
**Contribution:** 2
**Rating:** 5
**Confidence:** 3

**Summary:**

This paper proposes the Multimodal Academic Cover (MAC) benchmark, a new multimodal evaluation framework to assess the ability of Large Multimodal Models (LMMs) in both understanding and generating academic journal covers. MAC includes two core tasks:
1) Image2Text: Generating cover stories from given journal cover images and articles, assessing how well LMMs understand and articulate the scientific concepts depicted. 2) Text2Image: Generating academic journal cover images from cover stories and articles, which tests the models' capability to visualize complex scientific concepts in a visually compelling and contextually accurate manner. Additionally, the paper introduces Multimodal Agent Linkage (MAL), a technique that combines LLMs (e.g., ChatGPT) with LMMs to improve their performance in handling long-context, academic content. MAL enhances LMMs by leveraging LLMs’ strengths in processing complex scientific texts, translating them into prompts or descriptions that are suitable for image generation.

**Strengths:**

1. This paper’s structure is clear.
2. This paper is well-written.

**Weaknesses:**

1. There are too few MLLMS for comparison, and some typical MLLMS are not included, such as Emu2, Flamingo, etc.
2. Research on related work is inadequate. Some other work that included both human-involved and Large Model-involved evaluations is not compared in the related work.
3. The data set is too small. With only 5872 journal issues covered, it seems likely that the model will be trained on this basis to overfit, and diversity seems difficult to guarantee.
4. The distribution of the data set appears to be very imbalanced. There is also a lack of more detailed analysis. As Table 4 shows, for example, biology has many times the number of issues.

**Questions:**

Please refer to the weaknesses.

---

> ### Author Response · Authors · 2024-11-24
>
> Thank you for your thoughtful feedback and for recognizing the strengths of our work. We truly appreciate your insights.
> >This paper’s structure is clear.
>
> >This paper is well-written.
>
> Below, please find our detailed responses to your concerns.
>
> ## More Models
>
> >There are too few MLLMS for comparison, and some typical MLLMS are not included, such as Emu2, Flamingo, etc.
>
> Thank you for your feedback. In our study, we compared a selection of the most advanced MLLMs available at the time, including both commercial and open-source models such as DALL·E 3, GPT-4V, Gemini, CogView-3, GLM-4V, LLaVA, LLaMA-adapter, and MiniGPT4. We acknowledge that some notable models, such as Emu2 and Flamingo, were not included. While it is challenging to cover all MLLMs in a single study, we appreciate your suggestion and will make an effort to incorporate these models in future experiments to further enhance the comprehensiveness of our comparisons.
>
> ## Related Work
>
> >Research on related work is inadequate. Some other work that included both human-involved and Large Model-involved evaluations is not compared in the related work.
>
> Thank you for pointing this out. We have mentioned some papers such as MBench (Liu et al., 2023d), CLAIR (Chan et al.,
> 2023), and DEsignBench (Lin et al., 2023). If you have specific articles or studies in mind that you think should be cited, we would greatly appreciate your guidance and will make sure to include the appropriate references as per your suggestions.
>
> ## Dataset Size
>
> >The data set is too small. With only 5872 journal issues covered, it seems likely that the model will be trained on this basis to overfit, and diversity seems difficult to guarantee.
>
> Thank you for your insightful feedback. We acknowledge that the dataset is relatively small. However, we would like to clarify that our primary intention is not to use this dataset for model training but rather as a specialized testing benchmark. This approach minimizes the risk of overfitting during evaluation.
>
> Regarding diversity, we continuously update the dataset with newly published journal covers, ensuring that the test set remains fresh and reflective of a broad spectrum of styles and categories. This dynamic nature not only helps maintain diversity but also prevents data leakage, similar to the way benchmarks like CASP operate in the protein modeling domain. We hope this addresses your concerns and illustrates the unique role this dataset is designed to play.
>
> ## Imbalanced Disciplines
>
> >The distribution of the data set appears to be very imbalanced. There is also a lack of more detailed analysis. As Table 4 shows, for example, biology has many times the number of issues.
>
> Thank you for pointing this out. You are correct that the dataset distribution is imbalanced, with fields like biology having significantly more issues, as shown in Table 4. This imbalance reflects the real-world distribution of journal publications, where some disciplines, such as biology, produce a larger volume of content than others. While this may introduce challenges, we believe it also enhances the realism of the dataset, making it more representative of the scenarios in which models may be deployed.

---

### Official Review · Reviewer_uojW · 2024-11-09

**Soundness:** 3
**Presentation:** 3
**Contribution:** 2
**Rating:** 5
**Confidence:** 3

**Summary:**

This paper presents MAC, a benchmark for journal cover generation and understanding. After constructing the benchmark, the authors conduct extensive experiments with many LMMs like GPT-4V and LLaVA. The authors also propose a new method to improve long-context understanding.

**Strengths:**

- The proposed benchmark is useful for specific fields like journal and advertisement industry.
- Many popular models are tested, which is good.

**Weaknesses:**

- The benchmark only includes 5K images, which is pretty small. Besides, I am not sure how much impact would this benchmark have in a broader aspect. It seems the journal cover is only useful for some specific business.
- It is a task of image generation and understanding with LLMs, so methods that advance both should be discussed such as Emu, GILL, DreamLLM, SEED, and VILA-U.
- I think LMM is not a common expression. I suggest authors to use MLLM instead (multimodal LLM).

**Questions:**

n/a

---

> ### Author Response · Authors · 2024-11-24
>
> Thank you for your thoughtful feedback and for recognizing the strengths of our work. We truly appreciate your insights.
> >The proposed benchmark is useful for specific fields like journal and advertisement industry.
>
> >Many popular models are tested, which is good.
>
> Below, please find our detailed responses to your concerns.
> ## Dataset Size
> >The benchmark only includes 5K images, which is pretty small. Besides, I am not sure how much impact would this benchmark have in a broader aspect. It seems the journal cover is only useful for some specific business.
>
> Thank you for your thoughtful feedback. We agree that the MAC dataset is relatively small, which makes it less suitable as a pretraining resource. Instead, our primary focus in this work is to position MAC as a specialized testing benchmark. Additionally, we would like to highlight that the MAC dataset is designed to be continually updated as new journal covers are published annually. This dynamic updating process ensures that the testing set evolves over time, helping to prevent data leakage at its root. This approach is somewhat analogous to how the CASP benchmark operates in the protein modeling domain, ensuring the long-term relevance and utility of MAC for evaluating related tasks.
>
> ## More Models
>
> >It is a task of image generation and understanding with LLMs, so methods that advance both should be discussed such as Emu, GILL, DreamLLM, SEED, and VILA-U.
>
> Thank you for your valuable suggestions. In our study, we have already evaluated several of the most advanced multimodal models, including both commercial and open-source systems such as DALL·E 3, GPT-4V, Gemini, CogView-3, GLM-4V, LLaVA, LLaMA-adapter, and MiniGPT4. The models you mentioned, such as Emu, GILL, DreamLLM, SEED, and VILA-U, are indeed promising and worth testing. However, given the breadth of existing multimodal models, it is challenging to cover them all within the scope of a single study. We appreciate your suggestions and will consider including these models in future experiments to further enhance the comprehensiveness of our benchmarks.
>
> ## MLLM
>
> >I think LMM is not a common expression. I suggest authors to use MLLM instead (multimodal LLM).
>
> Thank you for the suggestion. We will replace “LMM” with “MLLM” to align with more commonly used terminology.

---

### Meta-Review · Area_Chair_V6uc · 2024-12-19

**Metareview:**

This paper presents a benchmark of journal covers, and uses it to evaluate both journal cover generation (T2I) and understanding (I2T).

The strengths of this paper are: 1) this journal cover data set is new, probably the first data set focusing on journal cover.

However, the paper suffers from several major issues: 1) The scope of the data set is too narrow, only journal covers and mainly from three journals. It will be more interesting if the data set contains more diverse covers or even infographics. 2)  Many MLLM methods are not compared, and moreover, reviewers also pointed out some related works are missing.

The topic is quite specific and may only be interesting to a small group of audience, and the overall contribution of this paper is not high (a small data set with limited scope). So I will recommend "reject".

**Additional Comments On Reviewer Discussion:**

Reviewer asked questions about data set size, data set scope, more MLLM models for testing, data leakage (i.e., these covers from famous journals may already be seen by MLLM), and the authors provided rebuttal to explain that the relatively data set size is for evaluation only, data set scope is a design choice, not possible to cover all MLLM models given there are so many, and some of most recent journal covers in the data sets should not be seen by MLLM etc.

I understood the points made in the rebuttal by the authors, but it does not change the major weakness of the paper: the data set is about a specific topic without broad interest, and the contributions of the paper is not high enough, given the data set size and scope.

---

### Decision · Program_Chairs · 2025-01-22

Reject